# Application of a life table approach to assess duration of BNT162b2 vaccine-derived immunity by age using COVID-19 case surveillance data during the Omicron variant period

Maya R. Sternberg[1]*, Amelia Johnson[1], Justice King[1], Akilah R. Ali[1], Lauren Linde[1], Abiola O. Awofeso[2], Jodee S. Baker[3], Nagla S. Bayoumi[4], Steven Broadway[5], Katherine Busen[6], Carolyn Chang[7], Iris Cheng[8], Mike Cima[9], Abi Collingwood[3], Vajeera Dorabawila[10], Cherie Drenzek[11], Aaron Fleischauer[1], Ashley Gent[5], Amanda Hartley[12], Liam Hicks[13], Mikhail Hoskins[14], Amanda Jara[11], Amanda Jones[1], Saadiah I. Khan[4], Ishrat Kamal-Ahmed[15], Sarah Kangas[16], FNU Kanishka[15], Alison Kleppinger[17], Anna Kocharian[16], Tomás M. León[18], Ruth Link-Gelles[1], B. Casey Lyons[1], John Masarik[2], Andrea May[19], Donald McCormick[9], Stephanie Meyer[20], Lauren Milroy[21], Keeley J. Morris[20], Lauren Nelson[18], Enaholo Omoike[6], Komal Patel[11], Michael Pietrowski[22], Melissa A. Pike[23], Tamara Pilishvili[1], Xandy Peterson Pompa[13], Charles Powell[17], Kevin Praetorius[24], Eli Rosenberg[10], Adam Schiller[1], Mayra L. Smith-Coronado[23], Emma Stanislawski[25], Kyle Strand[15], Buddhi P. Tilakaratne[2], Hailey Vest[21], Caleb Wiedeman[12], Allison Zaldivar[19], Benjamin Silk[1], Heather M. Scobie[1]

1 COVID-19 Response, Centers for Disease Control and Prevention, Atlanta, Georgia, United States of America, 2 Community Health Administration, DC Department of Health, Washington, District of Columbia, United States of America, 3 Division of Population Health, Utah Department of Health and Human Services, Salt Lake City, Utah, United States of America, 4 Communicable Disease Service, New Jersey Department of Health, Trenton, New Jersey, United States of America, 5 Division of Disease Control and Health Protection, Florida Department of Health, Tallahassee, Florida, United States of America, 6 Division of Communicable Disease, Michigan Department of Health and Human Services, Lansing, Michigan, United States of America, 7 Communicable Disease Service, New York City Department of Health and Mental Hygiene, Long Island City, New York, United States of America, 8 Bureau of Immunization, New York City Department of Health and Mental Hygiene, Long Island City, New York, United States of America, 9 Epidemilogy, Arkansas Department of Health, Little Rock, Arkansas, United States of America, 10 Bureau of Surveillance and Data Systems, Division of Epidemiology, Albany, New York State Department of Health, New York, NY, United States of America, 11 Acute Epidemiology, Georgia Department of Public Health, Atlanta, Georgia, United States of America, 12 Communicable and Environmental Diseases and Emergency Preparedness, Nashville, Tennessee Department of Health, Nashville, Tennessee, United States of America, 13 Bureau of Infectious Disease and Services, Arizona Department of Health Services, Phoenix, Arizona, United States of America, 14 Communicable Disease, North Carolina Department of Health and Human Services, Raleigh, North Carolina, United States of America, 15 Division of Public Health, Nebraska Department of Health and Human Services, Lincoln, Nebraska, United States of America, 16 COVID-19 Data and Surveillance Unit, Wisconsin Department of Health Services, Madison, Wisconsin, United States of America, 17 Epidemiology and Infectious Disease Section, Connecticut Department of Public Health, Hartford, Connecticut, United States of America, 18 Center for Infectious Diseases, California Department of Public Health, Sacramento, California, United States of America, 19 Bureau of Epidemiology and Public Health Informatics, Kansas Department of Health and Environment, Kansas, Missouri, United States of America, 20 Infectious Disease Epidemiology, Prevention and Control Division, Minnesota Department of Health, Saint Paul, Minnesota, United States of America, 21 Disease Epidemiology and Prevention Division, Indiana Department of Health, Indianapolis, Indiana, United States of America, 22 Division of Disease Control, Philadelphia Department of Public Health, Philadelphia, Pennsylvania, United States of America, 23 Disease Control and Public Health Response Division, Colorado Department of Public Health and Environment, Denver, Colorado, United States of America, 24 CDC Foundation, Atlanta, Georgia, United States of America, 25 Epidemiology and Response Division, New Mexico Department of Health, Santa Fe, New Mexico, United States of America

* mrs7@cdc.gov



**Data Availability Statement:** The data availability statement should be updated to say that all census data used for this study can be found at https://

www2.census.gov/programs-surveys/popest/tables/2010-2019/state/asrh/ the file name used is sc-est2019-alldata6.csv.

**Funding:** The authors received no specific funding for this work.

**Competing interests:** The authors have declared that no competing interests exist.

# Abstract

## Background

SARS-CoV-2 Omicron variants have the potential to impact vaccine effectiveness and duration of vaccine-derived immunity. We analyzed U.S. multi-jurisdictional COVID-19 vaccine breakthrough surveillance data to examine potential waning of protection against SARS-CoV-2 infection for the Pfizer-BioNTech (BNT162b) primary vaccination series by age.

## Methods

Weekly numbers of SARS-CoV-2 infections during January 16, 2022–May 28, 2022 were analyzed by age group from 22 U.S. jurisdictions that routinely linked COVID-19 case surveillance and immunization data. A life table approach incorporating line-listed and aggregated COVID-19 case datasets with vaccine administration and U.S. Census data was used to estimate hazard rates of SARS-CoV-2 infections, hazard rate ratios (HRR) and percent reductions in hazard rate comparing unvaccinated people to people vaccinated with a Pfizer-BioNTech primary series only, by age group and time since vaccination.

## Results

The percent reduction in hazard rates for persons 2 weeks after vaccination with a Pfizer-BioNTech primary series compared with unvaccinated persons was lowest among children aged 5–11 years at 35.5% (95% CI: 33.3%, 37.6%) compared to the older age groups, which ranged from 68.7%–89.6%. By 19 weeks after vaccination, all age groups showed decreases in the percent reduction in the hazard rates compared with unvaccinated people; with the largest declines observed among those aged 5–11 and 12–17 years and more modest declines observed among those 18 years and older.

## Conclusions

The decline in vaccine protection against SARS-CoV-2 infection observed in this study is consistent with other studies and demonstrates that national case surveillance data were useful for assessing early signals in age-specific waning of vaccine protection during the initial period of SARS-CoV-2 Omicron variant predominance. The potential for waning immunity during the Omicron period emphasizes the importance of continued monitoring and consideration of optimal timing and provision of booster doses in the future.

## Introduction

When COVID-19 vaccines were authorized and approved by the Food and Drug Administration (FDA) in the United States, the Centers for Disease Control and Prevention (CDC) began tracking cases, hospitalizations, and deaths among vaccinated persons [1]. The interpretability of this initial approach to COVID-19 vaccine breakthrough surveillance was hampered by limited reporting representativeness, a lack of denominators for rate calculations, and the absence of an unvaccinated comparison group. As state and local public health departments developed the capacity to routinely link case surveillance and immunization registry data, CDC collaborated with these jurisdictions to systematically monitor disease and mortality rates stratified by

vaccination status [2]. Multi-jurisdictional surveillance data showed that COVID-19 vaccines provided strong protection against COVID-19-associated hospitalization and death in a period when the Delta variant was predominant [2]. A study during the Delta period showed evidence of waning protection against infection but strong protection against death 6 months after vaccination [3, 4]; these findings were similar to other published studies from Israel and the United States [5–8]. More recently, case surveillance data have demonstrated continued protection from vaccination during Omicron, especially against severe outcomes [9, 10]. However, a study from New York state, which used data from linked statewide immunization and testing databases to compare vaccine effectiveness (VE) in children aged 5–11 years to those aged 12–17 years and vaccinated after the emergence of Omicron, demonstrated patterns that suggested rapid declines in VE in both age groups [11].

This analysis uses hazard rates of SARS-CoV-2 infection estimated from surveillance and vaccine administrative data to indirectly investigate the duration of COVID-19 vaccine protection over time and by age group. We applied life table methods during a period of SARS-CoV-2 Omicron variant predominance for persons 5 years and older who received the BNT162b2 (Pfizer-BioNTech) COVID-19 vaccine following authorization for children ages 5–11 years on October 29, 2021. Life tables are regularly used as a tool to study survival, incidence, and mortality in human populations and part of a suite of statistical methods that account for the special nature of time-to-event data called survival analysis.

## Materials methods

### COVID-19 case data by vaccination status

We analyzed reported numbers of SARS-CoV-2 infections by age group (5–11, 12–17, 18–49, 50–64, ≥65 years of age) from 22 U.S. jurisdictions (AR, AZ, CA, CO, CT, DC, FL, GA, IN, KS, MI, MA, MN, NC, NE, NJ, NM, NYC, PHL, TN, UT, WI); ~53% of the U.S. population) with routine linkages between COVID-19 case surveillance and immunization information system (IIS) data reported to CDC during January 16, 2022 –May 28, 2022 through two different mechanisms. Nine jurisdictions reported line-level data for all COVID-19 cases, including date of specimen collection, vaccine product, and dates of vaccination; 13 jurisdictions reported line-level data on cases among vaccinated persons, together with aggregated weekly counts of cases among partially vaccinated and unvaccinated persons by age group and epidemiological week of the SARS-COV-2 positive test [12, 13]. A vaccinated person was defined as a person with SARS-CoV-2 RNA or antigen detected in a respiratory specimen collected ≥14 days after verifiably completing the primary series of an FDA-authorized or approved COVID-19 vaccine. An unvaccinated person was defined as a person who was not verified to have received any COVID-19 vaccine doses before the positive SARS-CoV-2 specimen collection date. Analyses were limited to the period January 16, 2022–May 28, 2022 when the Omicron variant was predominant. We defined four vaccination cohort periods (1, 2, 3, and 4) based on MMWR weeks that roughly corresponded to calendar months: January 16–February 5, February 6–26, February 27–April 2, or April 3–30. Since Pfizer-BioNTech vaccine was the only vaccine authorized by the FDA for children during this period, we excluded individuals who received vaccinations other than the Pfizer-BioNTech vaccine; we also excluded those who received at least one FDA-authorized vaccine dose but did not complete a primary series ≥14 days before the positive specimen collection date. Based on earliest eligibility for first booster doses at 5 months after Pfizer-BioNTech primary series [14], persons newly vaccinated with a primary series and not moderately to severely immunocompromised would not have been able to receive a booster dose and reach ≥14 days after the booster dose (i.e., 22 weeks) within the period of follow-up for this study.

## Population data by vaccination status

Vaccine administration (coverage) data reported to CDC were aggregated by U.S. reporting jurisdiction, MMWR week of vaccination ($\geq$14 days after completing the primary vaccine series), FDA-approved vaccine products, and age group (5–11, 12–17, 18–49, 50–64, $\geq$65 years). To estimate the number of unvaccinated persons in each MMWR week, the 2019 U.S. Census population estimates by jurisdiction and age group were used (except for California, where State Department of Finance 2021 population projections were determined to be more accurate). These data were used to estimate the number of unvaccinated persons each MMWR week by subtracting the cumulative number of vaccinated (all products) and partially vaccinated persons (all products) from the respective population totals for each jurisdiction and age group. If the unvaccinated population was calculated to be less than 5% of the total census population for a given jurisdiction and age group, the unvaccinated population count was held fixed at 5% of the population total for that age group and jurisdiction. This continuity correction prevented the unvaccinated population from becoming unrealistically small due to potential overestimates of vaccination coverage [9]. This correction was applied to 5.3% of the combinations of 17 MMWR weeks, 7 age groups, and 22 jurisdictions in this study; primarily, estimates of the unvaccinated population for those 65 and older from five of the 22 jurisdictions were corrected. A sensitivity analysis (not shown) showed this continuity correction to not have a substantive impact on the reported findings in this paper.

## Ethics

The Office of the Associate Director for Science at the U.S. Centers for Disease Control and Prevention (CDC) determined the project to not to be human subjects' research. This study was approved as a public health surveillance activity and conducted consistent with applicable federal law and CDC policy [15]. Because these data were a secondary analysis of reported surveillance data that did not include any identifiable personal information, consent was not required. Funded by the Centers for Disease Control and Prevention.

## Application of life tables to breakthrough surveillance data

We conducted a period cohort life table analysis to estimate separate hazard functions for SARS-CoV-2 infection over weekly time intervals (in calendar time) for each vaccination cohort, as described elsewhere [3]. This approach allows the vaccination cohorts and outcome period to overlap, while removing those no longer at risk from the denominator (i.e., open cohort), which is an improvement over previously published closed cohort approach designs [5, 7]. Exposure time is measured from the date a person reaches $\geq$14 days after completing primary vaccination to the date of a SARS-CoV-2 positive test or the end of the study period (i.e., right censored). While a key assumption for methods of survival analysis is non-informative censoring, we used a fixed time censoring design, which has been proven to adhere to this assumption [16]. In our study, if a person was not reported as a case by May 28th, 2022 they were considered right censored. Additionally, it was not possible to account for any migrations in or out of the jurisdictions in this analysis.

The unvaccinated population was used as a reference group when interpreting COVID-19 hazard rates among vaccinated persons. Using hazard rates estimated from a life tables approach, the hazard rate ratio of the unvaccinated to the vaccinated population can be used to approximate crude VE (unadjusted for confounders). The relative hazard of a SARS-CoV-2 infection of the unvaccinated to the vaccinated was estimated by $HR(t) = \frac{h_U(t)}{h_V^i(t)}$, where $h_U(t)$ and $h_V^i(t)$ are the COVID-19 hazard rate at time $t$ for an unvaccinated cohort and vaccinated

cohort $i$, respectively. The percent reduction in hazard rate of SARS-CoV-2 infection among the vaccinated as compared to the unvaccinated was computed as $100\% \times \left(1 - \frac{1}{HR(t)}\right)$, which is a measure of crude VE. SAS 9.4 (SAS Institute, Cary, NC) was used to develop the algorithms to estimate the hazard rates among the vaccinated and unvaccinated, standard errors, and corresponding 95% confidence intervals [17].

To estimate the impact of waning immunity, we pooled across weekly vaccination cohorts where vaccination can occur at any time during January 16, 2022–May 28, 2022 such that the hazard was calculated as a function of time since vaccination (S1 Fig and S1 Table). There was no date of vaccination to use when constructing a reference group of the unvaccinated. Instead, the unvaccinated hazard rates from the same calendar weeks as the pooled vaccinated cohorts were standardized using the person-time distribution of the vaccinated as weights, similar to an analysis by Dorabawila et al. [11]. In this way, the standardized measure can be interpreted as percent reduction in the relative hazard rates among the vaccinated population, compared with the hazard rate among the unvaccinated population. We estimated the weekly percent reduction from 2 to 19 weeks after completing the primary series to minimize the potential chance for receipt of a booster dose at 20 weeks. The standard errors used to compute the 95% CI for the standardized HR and VE have been previously described [11, 18].

### Sensitivity analysis to explore the impact of prior infection among the unvaccinated

Because we are unable to directly account for immunity from prior SARS-CoV-2 infection using our surveillance data, we incorporated external data from repeated seroprevalence studies to adjust for the prevalence of prior infection among the unvaccinated as part of a sensitivity analysis [19]. During December 2021–February 2022, U.S. seroprevalence of infection increased overall from 29% in September 2021 to 34% in December 2021 to 43% in January 2022, with greater increases in seroprevalence observed in younger age groups [19]. Since there are no seroprevalence data available by vaccination status, we assumed that infections preferentially occurred in unvaccinated people during periods after the vaccine had been authorized. Estimates of the potential difference in protection from prior infection between unvaccinated and vaccinated people were calculated based on the age-group-specific differences in SARS-CoV-2 seroprevalence estimates September 2021 versus January 2022 for ages ≥12 years (17% for 12–17 years, 15% for 18–49 years, 12% for 50–63 years, and 7% for 65+ years); and between November 2021 to January 2022 for 5–11 years (16%) [19] (time period limited because of later vaccine authorization for this age group). To illustrate the potential impact of infection-derived immunity among the unvaccinated, we used these differences in seroprevalence estimates to remove infected persons from the total number unvaccinated at the beginning of the study period (week of January 16, 2021); this adjustment seemed clearer conceptually than also removing people from the vaccinated group who were already theoretically protected.

### Results

Across 22 reporting jurisdictions, there were 4,409,632 persons ≥5 years old who were vaccinated but had not received a booster dose during January 16 to May 28, 2022 (Table 1). Since children were the last group to receive FDA authorization for COVID-19 vaccination, the majority of those vaccinated without a booster during this period were children ages 5–11 years (42%). There were 1,387,186 (34.4%), 1,203,904 (29.9%), 963,880 (23.9%), and 473,756 (11.8%) vaccinated persons during each of the four cohorts: (1) January 16–February 5, (2) February 6–26, (3) February 27–April 2, or (4) April 3–30, respectively. During January 16 to

**Table 1. Number and rate of COVID-19 cases among persons ≥5 years old vaccinated with a BNT162b2 (Pfizer-BioNTech) primary series by age group—22 U.S. jurisdictions, January 16 to May 28, 2022ᵃ.**

|  | Number vaccinated persons | % | Number of COVID-19 cases among vaccinated persons | COVID-19 case rate per 100,000 vaccinated persons |
|---|---|---|---|---|
| **Total** | 4,409,632 | 100 | 45,911 | 1,041 |
| **Age group (years)** |  |  |  |  |
| **5–11** | 1,831,745 | 42% | 25,358 | 1,384 |
| **12–17** | 587,572 | 13% | 4,883 | 831 |
| **18–29** | 585,804 | 13% | 5,284 | 902 |
| **30–49** | 681,158 | 15% | 6,232 | 915 |
| **50–64** | 385,944 | 9% | 2,605 | 675 |
| **≥65** | 337,409 | 8% | 1,549 | 459 |

ᵃA vaccinated person was defined as a person with SARS-CoV-2 RNA or antigen detected in a respiratory specimen collected ≥14 days after verifiably completing the primary series of an FDA-authorized or approved COVID-19 vaccine.

May 28, 2022, a total of 1,041 COVID-19 cases were reported per 100,000 vaccinated persons aged ≥5 years overall. The highest case rate was among children ages 5–11 years (1,384 cases per 100,000 vaccinated persons).

Using the open cohort life table analysis, COVID-19 hazard rates were estimated for people in the unvaccinated and vaccinated cohorts in each age group during January 16 to May 28, 2022 (Fig 1). For age groups ≥12 years, the hazard functions in the unvaccinated cohort were

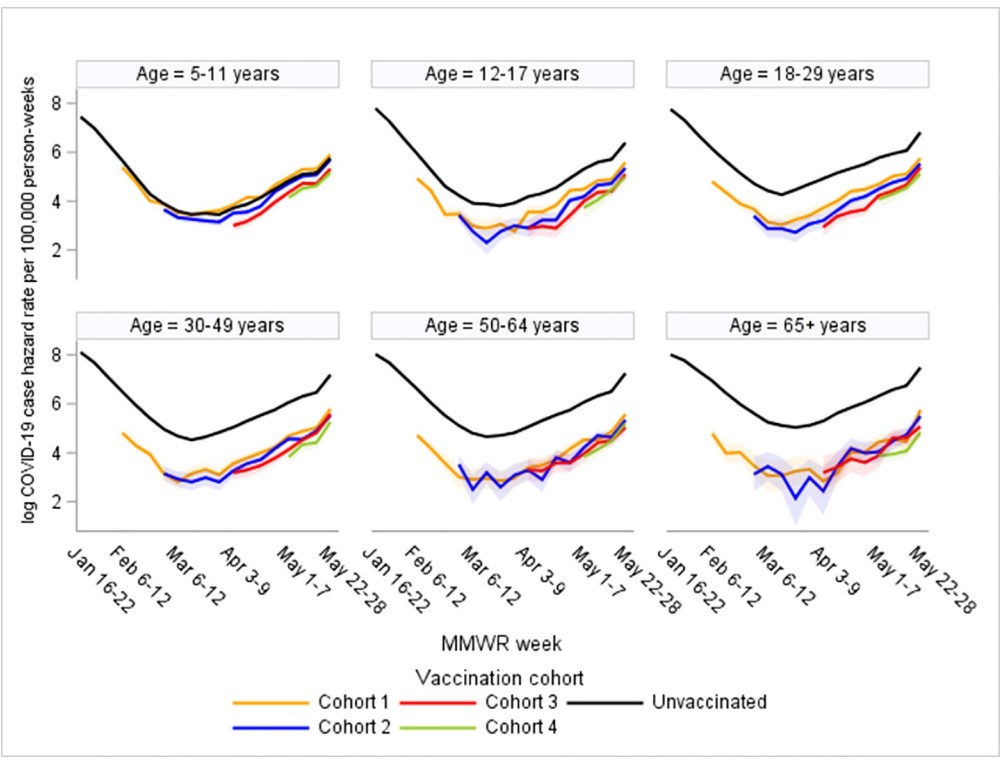

**Fig 1. Weekly trends in log COVID-19 hazard rates—22 U.S. jurisdictions, January 16 to May 28, 2022.** Log hazard rates (with 95% confidence intervals) among unvaccinated persons compared persons vaccinated with a Pfizer-BioNTech primary series by vaccination cohort and age group. Vaccination cohorts 1–4 reached ≥14 days after vaccination with a Pfizer-BioNTech primary series during: January 16–February 5, February 6–26, February 27–April 2, April 3–30, 2022, respectively.

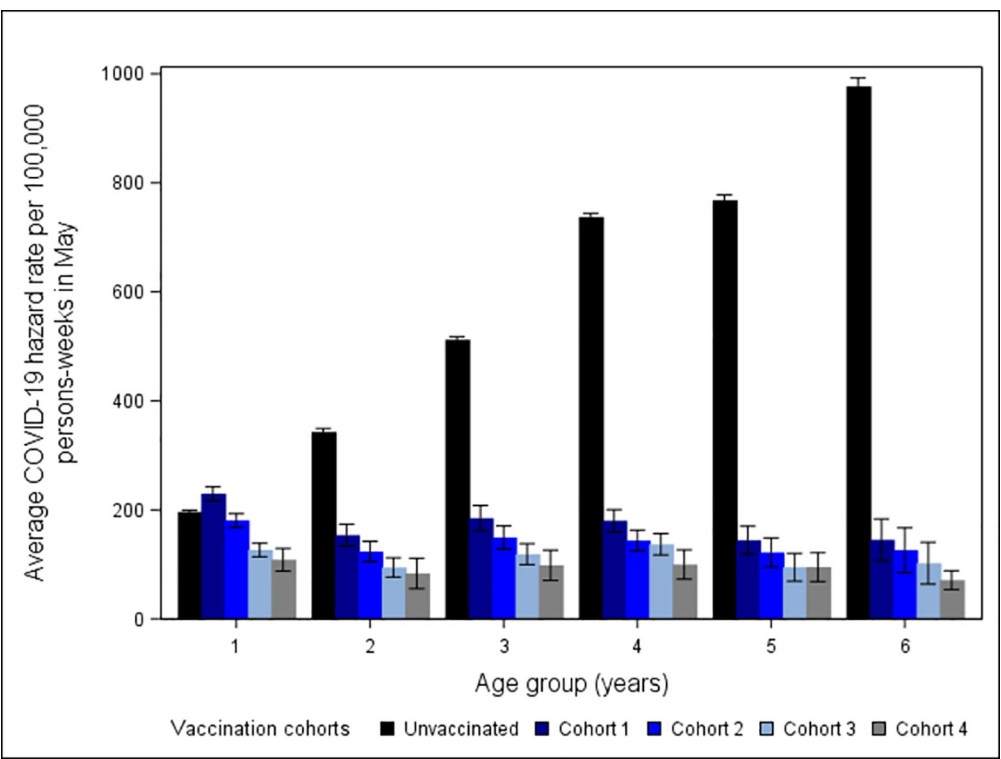

**Fig 2. COVID-19 hazard rates and 95% confidence intervals—22 U.S. jurisdictions, May 1–28, 2022.** Vaccination cohorts 1–4 reached ≥14 days after vaccination with a Pfizer-BioNTech primary series during: January 16–February 5, February 6–26, February 27–April 2, April 3–30, 2022, respectively.

approximately proportional to the hazard functions in the vaccinated cohorts during January–May 2022. However, children aged 5–11 years had the smallest difference in the hazard rates between the unvaccinated and vaccinated cohorts.

During May 1–28, 2022, higher COVID-19 hazard rates were observed in a stair-step pattern in cohorts that were vaccinated earlier with similar trends across all age groups, which is suggestive of waning vaccine protection (Fig 2). Children ages 5–11 years had the largest difference across the four vaccination cohorts, with the clearest distinction between the cohort vaccinated during January 16–February 5 (cohort 1) and the cohort vaccinated during April 3–30 (cohort 4). Unlike the older age groups, unvaccinated children ages 5–11 years had a lower COVID-19 hazard rate than vaccinated children in cohort 1 of the same age group.

During January 16 to May 28, 2022, all age groups had declines in the standardized COVID-19 case HRRs in unvaccinated persons compared with vaccinated persons, which corresponded to declines in the percent reductions in relative hazard rates between vaccinated and unvaccinated persons from 2 to 19 weeks after vaccination (Table 2, Fig 3, S2 and S3 Figs). From 2 to 19 weeks since vaccination, the percent reductions in hazard rates among vaccinated persons compared with unvaccinated persons decreased for persons aged 12–17 years from 68.7% (95% CI: 66.2, 71.1) to 52.3% (95% CI: 41.3, 61.2), for persons aged 18–29 years from 77.2% (95% CI: 75.2, 79.1) to 65.5% (95% CI: 57.8, 71.7), for persons aged 30–49 years from 82.9% (95% CI: 81.6, 84.1) to 73.3% (95% CI: 68.4, 77.4), for persons aged 50–64 years from 86.0% (95% CI: 84.4, 87.5) to 80.7% (95% CI: 74.7, 85.3), and for persons 65 years and older from 89.6% (95% CI: 88.2, 90.9) to 81.0% (95% CI: 72.9, 86.7). Children ages 5–11 years had a 35.5% (95% CI: 33.3, 37.6) percent reduction in COVID-19 hazard rate at 2 weeks after

**Table 2. Standardized hazard rate ratios (with 95% confidence intervals) for COVID-19 cases among unvaccinated persons compared with persons vaccinated with a Pfizer-BioNTech primary series and percent reduction in hazard rate among vaccinated persons compared with unvaccinated persons by age group and weeks since completing vaccination—22 U.S. jurisdictions, January 16 to May 28, 2022.**

| | Standardized Hazard Rate Ratios (95% CI) [a] | | | | | |
|---|---|---|---|---|---|---|
| | | Age group | | | | |
| Weeks since vaccination | 5–11 years | 12–17 years | 18–29 years | 30–49 years | 50–64 years | 65+ years |
| 2 | 1.55 (1.50, 1.60) | 3.2 (2.96, 3.46) | 4.39 (4.04, 4.78) | 5.85 (5.43, 6.30) | 7.16 (6.42, 7.98) | 9.62 (8.46, 10.95) |
| 3 | 1.40 (1.35, 1.47) | 3.17 (2.86, 3.51) | 4.27 (3.85, 4.73) | 5.59 (5.10, 6.13) | 7.84 (6.84, 8.99) | 9.89 (8.57, 11.41) |
| 4 | 1.30 (1.23, 1.37) | 2.60 (2.31, 2.92) | 4.32 (3.83, 4.88) | 5.74 (5.14, 6.42) | 7.27 (6.23, 8.47) | 13.26 (11.07, 15.89) |
| 5 | 1.37 (1.28, 1.46) | 2.98 (2.58, 3.44) | 4.75 (4.13, 5.46) | 5.81 (5.12, 6.58) | 8.98 (7.41, 10.9) | 12.31 (10.15, 14.92) |
| 6 | 1.37 (1.27, 1.48) | 3.25 (2.76, 3.81) | 4.57 (3.96, 5.28) | 6.29 (5.48, 7.22) | 6.23 (5.21, 7.46) | 11.15 (8.99, 13.83) |
| 7 | 1.43 (1.33, 1.55) | 3.32 (2.82, 3.91) | 5.07 (4.36, 5.9) | 7.09 (6.12, 8.22) | 6.70 (5.47, 8.20) | 9.99 (7.71, 12.95) |
| 8 | 1.32 (1.23, 1.42) | 3.54 (3.00, 4.17) | 4.30 (3.75, 4.92) | 6.47 (5.64, 7.42) | 7.06 (5.69, 8.75) | 9.67 (7.07, 13.24) |
| 9 | 1.32 (1.23, 1.41) | 3.34 (2.86, 3.89) | 4.07 (3.58, 4.62) | 5.21 (4.62, 5.86) | 7.03 (5.71, 8.65) | 7.99 (6.02, 10.60) |
| 10 | 1.28 (1.2, 1.37) | 3.04 (2.64, 3.50) | 4.41 (3.87, 5.01) | 5.16 (4.61, 5.79) | 7.92 (6.41, 9.78) | 8.38 (6.31, 11.12) |
| 11 | 1.23 (1.16, 1.31) | 3.30 (2.87, 3.79) | 4.07 (3.61, 4.58) | 5.27 (4.72, 5.88) | 6.76 (5.61, 8.15) | 8.32 (6.34, 10.93) |
| 12 | 1.08 (1.02, 1.14) | 2.88 (2.54, 3.27) | 3.33 (2.99, 3.71) | 4.92 (4.43, 5.46) | 6.89 (5.74, 8.26) | 8.84 (6.73, 11.60) |
| 13 | 1.07 (1.02, 1.13) | 2.39 (2.14, 2.67) | 3.19 (2.88, 3.55) | 4.65 (4.21, 5.14) | 5.87 (4.98, 6.91) | 8.01 (6.22, 10.31) |
| 14 | 1.00 (0.95, 1.05) | 2.42 (2.16, 2.71) | 3.30 (2.96, 3.68) | 4.90 (4.43, 5.43) | 5.77 (4.91, 6.77) | 7.96 (6.20, 10.24) |
| 15 | 0.88 (0.84, 0.92) | 2.46 (2.20, 2.76) | 2.93 (2.64, 3.26) | 4.65 (4.20, 5.15) | 6.44 (5.43, 7.64) | 7.04 (5.54, 8.94) |
| 16 | 0.92 (0.87, 0.97) | 2.32 (2.06, 2.61) | 2.94 (2.62, 3.29) | 4.30 (3.88, 4.76) | 5.55 (4.71, 6.55) | 7.11 (5.54, 9.12) |
| 17 | 0.88 (0.83, 0.93) | 2.06 (1.82, 2.33) | 2.60 (2.31, 2.94) | 4.31 (3.84, 4.83) | 5.11 (4.30, 6.07) | 8.75 (6.48, 11.79) |
| 18 | 0.87 (0.81, 0.92) | 2.21 (1.90, 2.58) | 2.60 (2.26, 2.99) | 4.02 (3.54, 4.57) | 4.89 (4.03, 5.93) | 7.12 (5.24, 9.67) |
| 19 | 0.75 (0.69, 0.81) | 2.09 (1.70, 2.57) | 2.90 (2.37, 3.54) | 3.74 (3.16, 4.43) | 5.18 (3.96, 6.79) | 5.27 (3.68, 7.54) |
| | Percent Reduction in Hazard Rate (95% CI) | | | | | |
| Weeks since vaccination | 5–11 years | 12–17 years | 18–29 years | 30–49 years | 50–64 years | 65+ years |
| 2 | 35.5 (33.3, 37.6) | 68.7 (66.2, 71.1) | 77.2 (75.2, 79.1) | 82.9 (81.6, 84.1) | 86.0 (84.4, 87.5) | 89.6 (88.2, 90.9) |
| 3 | 28.8 (25.7, 31.7) | 68.5 (65.0, 71.5) | 76.6 (74.0, 78.9) | 82.1 (80.4, 83.7) | 87.2 (85.4, 88.9) | 89.9 (88.3, 91.2) |
| 4 | 22.8 (18.6, 26.8) | 61.5 (56.8, 65.7) | 76.9 (73.9, 79.5) | 82.6 (80.5, 84.4) | 86.2 (84.0, 88.2) | 92.5 (91.0, 93.7) |
| 5 | 26.9 (21.9, 31.6) | 66.4 (61.2, 70.9) | 78.9 (75.8, 81.7) | 82.8 (80.5, 84.8) | 88.9 (86.5, 90.8) | 91.9 (90.2, 93.3) |
| 6 | 27.1 (21.5, 32.3) | 69.2 (63.8, 73.8) | 78.1 (74.8, 81.1) | 84.1 (81.7, 86.1) | 84 (80.8, 86.6) | 91.0 (88.9, 92.8) |
| 7 | 30.3 (24.6, 35.5) | 69.9 (64.5, 74.4) | 80.3 (77.1, 83.0) | 85.9 (83.7, 87.8) | 85.1 (81.7, 87.8) | 90.0 (87.0, 92.3) |
| 8 | 24.4 (18.6, 29.8) | 71.7 (66.6, 76.0) | 76.7 (73.3, 79.7) | 84.5 (82.3, 86.5) | 85.8 (82.4, 88.6) | 89.7 (85.8, 92.4) |
| 9 | 24.1 (18.5, 29.3) | 70.0 (65.1, 74.3) | 75.4 (72.0, 78.4) | 80.8 (78.4, 82.9) | 85.8 (82.5, 88.4) | 87.5 (83.4, 90.6) |
| 10 | 22.1 (16.7, 27.1) | 67.1 (62.2, 71.4) | 77.3 (74.2, 80.1) | 80.6 (78.3, 82.7) | 87.4 (84.4, 89.8) | 88.1 (84.2, 91.0) |
| 11 | 18.7 (13.5, 23.6) | 69.7 (65.2, 73.6) | 75.4 (72.3, 78.2) | 81.0 (78.8, 83.0) | 85.2 (82.2, 87.7) | 88.0 (84.2, 90.8) |
| 12 | 7.0 (1.7, 12.0) | 65.3 (60.7, 69.4) | 70.0 (66.6, 73.0) | 79.7 (77.4, 81.7) | 85.5 (82.6, 87.9) | 88.7 (85.1, 91.4) |
| 13 | 6.7 (1.6, 11.6) | 58.1 (53.2, 62.5) | 68.7 (65.2, 71.8) | 78.5 (76.3, 80.5) | 83.0 (79.9, 85.5) | 87.5 (83.9, 90.3) |
| 14 | -0.2 (-5.4, 4.8) | 58.7 (53.8, 63.0) | 69.7 (66.2, 72.8) | 79.6 (77.4, 81.6) | 82.7 (79.6, 85.2) | 87.4 (83.9, 90.2) |
| 15 | -13.5 (-19.1, -8.1) | 59.4 (54.4, 63.8) | 65.9 (62.1, 69.3) | 78.5 (76.2, 80.6) | 84.5 (81.6, 86.9) | 85.8 (81.9, 88.8) |
| 16 | -9.0 (-14.8, -3.5) | 56.9 (51.5, 61.6) | 66.0 (61.8, 69.6) | 76.7 (74.2, 79.0) | 82.0 (78.8, 84.7) | 85.9 (82.0, 89.0) |
| 17 | -13.5 (-20.1, -7.3) | 51.5 (45.2, 57.2) | 61.6 (56.6, 66.0) | 76.8 (74.0, 79.3) | 80.4 (76.7, 83.5) | 88.6 (84.6, 91.5) |
| 18 | -15.6 (-23.5, -8.3) | 54.8 (47.5, 61.2) | 61.5 (55.7, 66.5) | 75.1 (71.7, 78.1) | 79.5 (75.2, 83.1) | 85.9 (80.9, 89.7) |
| 19 | -34.2 (-45.7, -23.6) | 52.3 (41.3, 61.2) | 65.5 (57.8, 71.7) | 73.3 (68.4, 77.4) | 80.7 (74.7, 85.3) | 81.0 (72.9, 86.7) |

[a]Unvaccinated hazard rates from the same calendar weeks as the pooled vaccinated cohorts were standardized using the person-time distribution of the vaccinated as weights [11, 18].

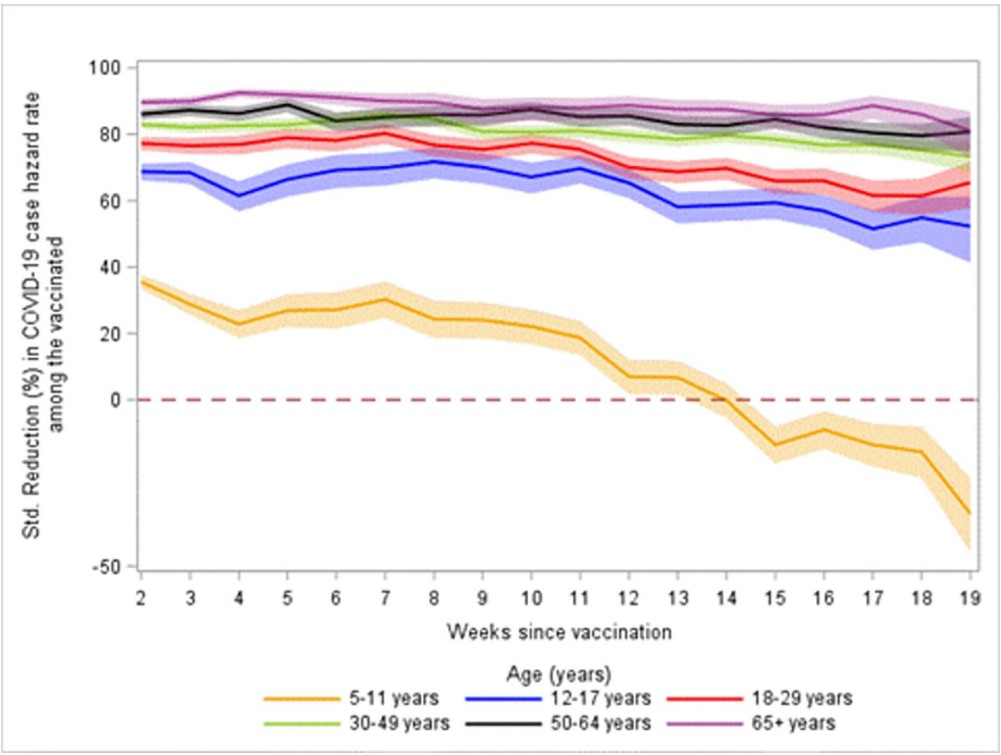

**Fig 3. Standardized percent reduction in hazard rate for COVID-19 cases—22 U.S. jurisdictions, January 16 to May 28, 2022.** Shaded area are 95% confidence intervals among persons vaccinated with a Pfizer-BioNTech primary series as compared to unvaccinated persons by age group and weeks since completing vaccination.

completing the primary series, but this percent reduction fell below 0% for children 5–11 years beginning at 14 weeks post-vaccination.

The sensitivity analysis, which evaluated the impact of differences in prior infection by vaccination status with a crude adjustment to the unvaccinated group, showed similar patterns as the original analysis, but with higher percent reductions in COVID-19 hazard rates among vaccinated persons compared with unvaccinated persons for all age groups (Fig 4, S4 and S5 Figs). From 2 to 19 weeks since vaccination, the percent reductions in hazard rates among vaccinated persons compared with unvaccinated persons for the sensitivity analysis changed from 78.8% (95% CI: 77.1, 80.4) to 68.1% (95% 60.9, 74.1) for persons 12–17 years, from 83.8% (95% CI: 82.4, 85.1) to 75.9% (95% CI: 70.5, 80.2) for persons aged 18–29, from 87.9% (95% CI: 87.0, 88.8) to 81.6% (95% CI: 78.2, 84.4) for persons aged 30–49, from 89.4% (95% CI: 88.2, 90.5) to 85.6% (95% CI: 81.1, 89.0) for persons aged 50–64, and from 91.2% (95% CI: 89.9, 92.2) to 84.0% (95% CI: 77.1, 88.8) for persons aged 65 years and older. Even after the adjustment, children ages 5–11 years showed more rapid decline from 55.0% (95% CI: 53.4, 56.4) to 7.3% (95% CI: -0.7, 14.6) at 2 to 19 weeks since vaccination, respectively (Fig 4).

## Discussion

This report describes the application of a life table analysis using COVID-19 case surveillance and vaccine administration data to evaluate the differences in hazard rates among people vaccinated and unvaccinated over time, which serves as a proxy of waning of vaccine protection against SARS-CoV-2 infection by age group during the Omicron period. At 19 weeks after vaccination, decreased protection against infection was observed for all age groups, residual

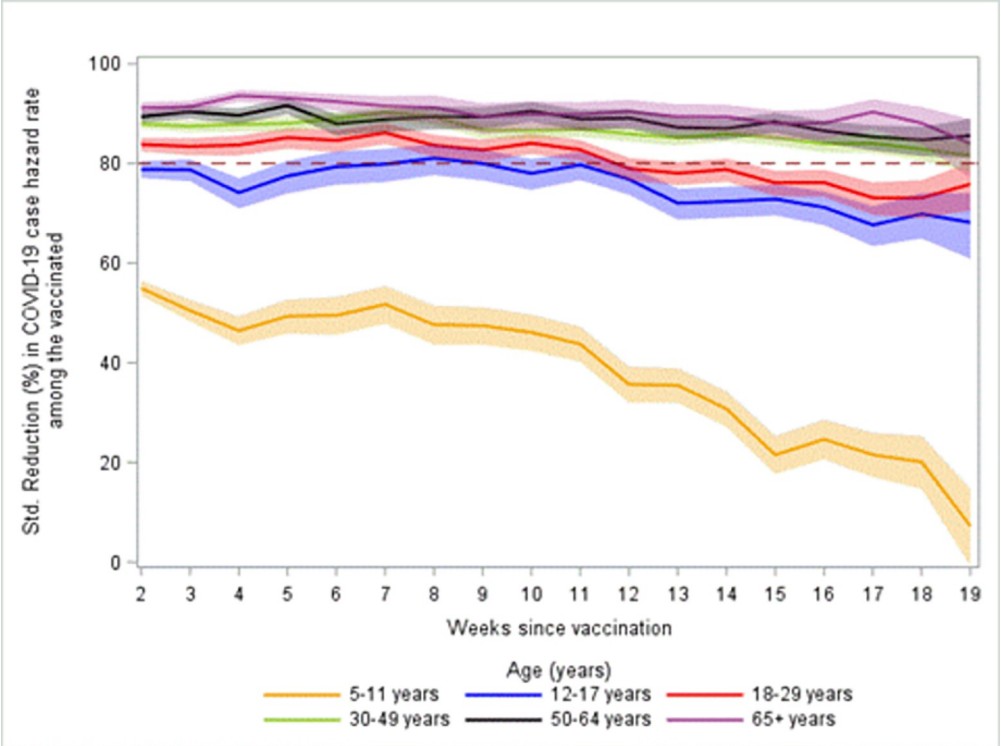

**Fig 4. Sensitivity analysis for standardized percent reduction in hazard rate for COVID-19 cases.** Adjusted by an estimate of differential prior infection based on seroprevalence data among the unvaccinated. Shaded area are 95% confidence intervals. A dashed reference line is plotted at a VE of 80%.

protection against infection was observed for ages 12 and older. Children aged 5–11 years had similar hazard rates among vaccinated and unvaccinated cohorts at 14 weeks (or at 19 weeks in the sensitivity analysis), indicating limited vaccine protection against infection after these time points. Vaccine protection against severe COVID-19 outcomes was not evaluated but has been shown elsewhere to be more robust and durable than protection against infection and symptomatic disease [3, 4, 20–22].

Our findings generally agree with trends noted in similar analyses of surveillance data published for New York state and North Carolina [6, 11, 23]. However, the apparent trend of more rapid waning immunity in children aged 5–11 years has not been observed in more robust VE studies that are able to more adequately control and adjust for epidemiologic confounding, though faster waning of immunity during the era of Omicron predominance has been noted for all age groups [22, 24–29]. A meta-review showed several other VE studies have shown residual protection against infection up to 5 months [23]. One VE study noted lower VE against infection for children ages 5–11 years (31%) compared with children ages 12–15 years (59%), but the difference was not statistically significant [30]. A test-negative case control study among children ages 5–11 years conducted during a similar period as our study also found VE against infection (20%) that waned after 3 months [31]. A hypothesis posed for faster waning of immunity in children 5–11 years of age was the use of one-third vaccine dosage in this younger group compared with children ages ≥12 years [11]; lower VE with increasing years of age within group (i.e., a presumed marker for body mass) has been observed in one investigation conducted among ages 5–11 years [32]. A likely explanation for the differences between the life table analyses of case surveillance data and well-designed VE studies are that

analyses of surveillance data have limited ability to control for covariates beyond analyses stratified by age, vaccine product, and time since vaccination. Analyses of case surveillance data are also generally unable to address testing and reporting biases, and differences in prior infection or prevention behaviors by vaccination status. Estimates of prior infection are higher in younger age groups and likely also differ by vaccination status [19], a potential bias that may explain the negative VE observed among children 5–11 years [33], and which we tried to demonstrate through a sensitivity analysis. Home-based antigen testing has increased over time, resulting in a greater number of unreported infections, which likely also differ by age and vaccination status [34, 35]. Higher seroprevalence has been observed among children 5–11 years, likely due in part to this being among the last groups to be authorized to receive the vaccine. Additionally, school-based testing practices are not well-documented but could have resulted in greater rates of detection of mild/asymptomatic cases in both vaccinated and unvaccinated children.

Our approach to comparing vaccinated and unvaccinated persons used case reporting and vaccine registry data that are useful for generating early signals to detect changes in rate ratios by vaccination status over time, which may indicate changes in protection due to emerging variants and waning immunity for evaluation with robust VE studies. Our methods build on similar analyses of case surveillance data published elsewhere [2, 3, 5–8, 11], and allow for overlapping vaccination cohorts to be evaluated over calendar time. However, integration of different monitoring and administration data poses challenges and is not as straightforward as analyses of cohort or test negative studies. Careful accounting is needed to capture the transitions from being unvaccinated to partially vaccinated and to completing a primary series, and from being uninfected to infected at any given week, which is difficult to assess accurately with imprecise data. The unvaccinated population must be crudely estimated using past census counts and vaccine administration data; by definition, this group lacks a natural time origin for comparison with the vaccinated population (i.e., a vaccination date). Also, age was calculated differently for the vaccinated (based on date of vaccination) and the unvaccinated (based on the date of the positive SARS-CoV-2 test) groups. The rationale for using age at vaccination for the analysis was that children 5–11 years old received only one-third of the Pfizer-BioNTech vaccine dose provided to children ≥12 years old [36]. Due to the short study period, very few people had an age group designation that was different at the time of vaccination and the time of their positive specimen collection.

This analysis had several limitations. Life tables estimated from aggregated data assume a closed demographic system, which is not typically accurate due to movement (e.g., an infected person reported in one jurisdiction may be vaccinated in a different jurisdiction) and could result in misclassification of vaccination status. We were unable to account for differences in underlying conditions, testing practices, prior infection, exposure risk, or prevention behaviors by vaccination status, age, geography, and over time; current results were influenced by the circulating Omicron variant, the relatively high prevalence of prior infection among children, and differences in testing practices and vaccine seeking behaviors compared to 2021. The vaccinated cohorts were limited to persons newly vaccinated during December 2021-January 2022; adults aged ≥18 years who were newly vaccinated after being eligible for more than a year may differ in ways that impact the generalizability of our findings. Variable linkage success for case surveillance and vaccination registry data also might have resulted in misclassifications that could influence estimates. This time to event analysis assumed that there was only one possible COVID-19 infection during the follow-up period, which was reasonable for newly vaccinated cohorts with a relatively short follow-up period after vaccination. However, reinfections (defined as occurring >90 days after prior infection) could not be excluded. In the future, reinfections could pose a limitation to this approach, especially during longer analysis

periods and periods of increasing incidence related to variants with enhanced transmissibility or immune escape. If so, the ability to identify reinfections through record linkage within surveillance data becomes more important and statistical methods for recurrent events could be applied. Lastly, the limitations of using direct age standardization have been well-described in the literature [18], but this approach serves as a reasonable solution to apply the unvaccinated rates and construct a synthetic reference group.

Despite its limitations, national case surveillance data were useful for assessing age-specific waning in vaccine protection during the initial period of SARS-CoV-2 Omicron variant predominance. With the anticipated emergence of new SARS-CoV-2 variants with increased immune escape and transmissibility, monitoring surveillance trends can provide useful signals for identifying unexpected changes in risk as the COVID-19 pandemic continues.

## Supporting information

**S1 Table. The number of weeks since vaccination is categorized using the duration between the week of vaccination to the week of infection.** The hazard is calculated as a function of time since vaccination by pooling across the weekly vaccination cohorts with the same time since vaccination (i.e. cohorts with the same color have same number of weeks since vaccination). This is an example from MMWR week 3 2022 to MMWR week 11 in 2022.
(DOCX)

**S2 Table. Completed STROBE checklist for observational study and analysis plan information.**
(DOCX)

**S1 Fig. Reconfigured schematic of the open vaccination cohorts defining the time origin from the date of vaccination.** The solid lines with arrows indicate subjects in vaccination who subsequently had a positive SARS-Cov-2 test at various points in calendar time. The dotted lines subjects in the vaccination cohort without an observed SARS-Cov-2 test (right-censored). Lines with the same color have the same duration since vaccination to an observed SARS-Cov-2 test. These subjects are pooled to display an analysis with a single common hazard function across the cohorts with time since vaccination as the x-axis.
(TIF)

**S2 Fig. Weekly trends in the percent reduction in hazard rate for COVID-19 cases (with 95% confidence intervals) among children 5–17 years vaccinated with a Pfizer-BioNTech primary series as compared to unvaccinated persons by age group and vaccination cohort—22 U.S. jurisdictions, January 16 to May 28, 2022. Vaccination cohorts 1–4 reached ≥14 days after vaccination with a Pfizer-BioNTech primary series during: January 16–February 5, February 6–26, February 27–April 2, April 3–30, 2022, respectively.** Black dashed reference line plotted at a VE of 80%.
(TIF)

**S3 Fig. Weekly trends in the percent reduction in hazard rate for COVID-19 cases (with 95% confidence intervals) among persons 18 years and older vaccinated with a Pfizer-BioNTech primary series as compared to unvaccinated persons by age group and vaccination cohort—22 U.S. jurisdictions, January 16 to May 28, 2022.** Vaccination cohorts 1–4 reached ≥14 days after vaccination with a Pfizer-BioNTech primary series during: January 16–February 5, February 6–26, February 27–April 2, April 3–30, 2022, respectively. Black dashed reference line plotted at a VE of 80%.
(TIF)

**S4 Fig. Sensitivity analysis of the weekly trends in the percent reduction in hazard rate for COVID-19 cases (with 95% confidence intervals) among children 5–17 years vaccinated with a Pfizer-BioNTech primary series as compared to unvaccinated persons with a crude adjustment of protection from prior infection by age group and vaccination cohort—22 U. S. jurisdictions, January 16 to May 28, 2022.** Vaccination cohorts 1–4 reached ≥14 days after vaccination with a Pfizer-BioNTech primary series during: January 16–February 5, February 6–26, February 27–April 2, April 3–30, 2022, respectively. For the sensitivity analysis, the total number unvaccinated at the beginning of the study period (week of January 16, 2021) was adjusted by a crude estimate of the potential difference in protection from prior infection by vaccination status using the difference in estimated SARS-CoV-2 seroprevalence by age group for September 2021 to January 2022 for ages 12–17 years and between November 2021 to January 2022 for 5–11 years [18], based on the later vaccine authorization for this age group. Black dashed reference line plotted at a VE of 80%.
(TIF)

**S5 Fig. Sensitivity analysis of the weekly trends in the percent reduction in hazard rate for COVID-19 cases (with 95% confidence intervals) among persons 18 years and older vaccinated with a Pfizer-BioNTech primary series as compared to unvaccinated persons with a crude adjustment of protection from prior infection by age group and vaccination cohort—22 U.S. jurisdictions, January 16 to May 28, 2022. Vaccination cohorts 1–4 reached ≥14 days after vaccination with a Pfizer-BioNTech primary series during: January 16–February 5, February 6–26, February 27–April 2, April 3–30, 2022, respectively.** For the sensitivity analysis, the total number unvaccinated at the beginning of the study period (week of January 16, 2021) was adjusted by a crude estimate of the potential difference in protection from prior infection by vaccination status using the difference in estimated SARS-CoV-2 seroprevalence by age group for September 2021 to January 2022 for ages ≥18 years [18]. Black dashed reference line plotted at a VE of 80%.
(TIF)

## Acknowledgments

We would like to acknowledge Aaron Bieringer, Miriam Muscoplat, Sydney Kuramoto, Amanda Markelz, Amy Saupe, Corinne Holtzman, Kathy Como-Sabetti, and Ruth Lynfield for their help in preparing advising on interpretation of these data.

**Disclaimer:** The views and opinions expressed by the authors are their own and do not necessarily represent the views and opinions of the CDC or those of any state health department or health services agency.

## Author Contributions

**Conceptualization:** Maya R. Sternberg, Amelia Johnson, Ruth Link-Gelles, Tamara Pilishvili, Eli Rosenberg, Benjamin Silk, Heather M. Scobie.

**Data curation:** Amelia Johnson, Justice King, Akilah R. Ali, Lauren Linde, Abiola O. Awofeso, Jodee S. Baker, Nagla S. Bayoumi, Steven Broadway, Katherine Busen, Carolyn Chang, Iris Cheng, Mike Cima, Abi Collingwood, Vajeera Dorabawila, Cherie Drenzek, Aaron Fleischauer, Ashley Gent, Amanda Hartley, Liam Hicks, Mikhail Hoskins, Amanda Jara, Amanda Jones, Saadiah I. Khan, Ishrat Kamal-Ahmed, Sarah Kangas, FNU Kanishka, Alison Kleppinger, Anna Kocharian, Tomás M. León, B. Casey Lyons, John Masarik, Andrea May, Donald McCormick, Stephanie Meyer, Lauren Milroy, Keeley J. Morris, Lauren Nelson, Enaholo Omoike, Komal Patel, Michael Pietrowski, Melissa A. Pike, Xandy Peterson

Pompa, Charles Powell, Kevin Praetorius, Eli Rosenberg, Adam Schiller, Mayra L. Smith-Coronado, Emma Stanislawski, Kyle Strand, Buddhi P. Tilakaratne, Hailey Vest, Caleb Wiedeman, Allison Zaldivar.

**Formal analysis:** Maya R. Sternberg, Amelia Johnson.

**Investigation:** Maya R. Sternberg, Amelia Johnson, Justice King, Akilah R. Ali, Lauren Linde, Jodee S. Baker, Nagla S. Bayoumi, Steven Broadway, Katherine Busen, Carolyn Chang, Iris Cheng, Mike Cima, Abi Collingwood, Cherie Drenzek, Aaron Fleischauer, Ashley Gent, Amanda Hartley, Liam Hicks, Mikhail Hoskins, Amanda Jara, Amanda Jones, Saadiah I. Khan, Ishrat Kamal-Ahmed, Sarah Kangas, FNU Kanishka, Alison Kleppinger, Anna Kocharian, Tomás M. León, Ruth Link-Gelles, B. Casey Lyons, John Masarik, Andrea May, Donald McCormick, Stephanie Meyer, Lauren Milroy, Keeley J. Morris, Lauren Nelson, Enaholo Omoike, Komal Patel, Michael Pietrowski, Melissa A. Pike, Tamara Pilishvili, Xandy Peterson Pompa, Charles Powell, Kevin Praetorius, Eli Rosenberg, Adam Schiller, Mayra L. Smith-Coronado, Emma Stanislawski, Kyle Strand, Buddhi P. Tilakaratne, Hailey Vest, Caleb Wiedeman, Allison Zaldivar, Benjamin Silk, Heather M. Scobie.

**Methodology:** Maya R. Sternberg, Amelia Johnson, Tamara Pilishvili, Heather M. Scobie.

**Project administration:** Ruth Link-Gelles, Benjamin Silk, Heather M. Scobie.

**Resources:** Amelia Johnson, Lauren Linde, Abiola O. Awofeso, Jodee S. Baker, Nagla S. Bayoumi, Steven Broadway, Katherine Busen, Carolyn Chang, Iris Cheng, Mike Cima, Abi Collingwood, Vajeera Dorabawila, Cherie Drenzek, Aaron Fleischauer, Ashley Gent, Amanda Hartley, Liam Hicks, Mikhail Hoskins, Amanda Jara, Amanda Jones, Saadiah I. Khan, Ishrat Kamal-Ahmed, Sarah Kangas, FNU Kanishka, Alison Kleppinger, Anna Kocharian, Tomás M. León, B. Casey Lyons, John Masarik, Andrea May, Donald McCormick, Stephanie Meyer, Lauren Milroy, Keeley J. Morris, Lauren Nelson, Enaholo Omoike, Komal Patel, Michael Pietrowski, Melissa A. Pike, Xandy Peterson Pompa, Charles Powell, Kevin Praetorius, Eli Rosenberg, Adam Schiller, Mayra L. Smith-Coronado, Emma Stanislawski, Kyle Strand, Buddhi P. Tilakaratne, Hailey Vest, Caleb Wiedeman, Allison Zaldivar.

**Software:** Maya R. Sternberg, Amelia Johnson.

**Supervision:** Benjamin Silk, Heather M. Scobie.

**Validation:** Amelia Johnson, Justice King, Akilah R. Ali, Lauren Linde, Jodee S. Baker, Nagla S. Bayoumi, Steven Broadway, Katherine Busen, Carolyn Chang, Iris Cheng, Mike Cima, Abi Collingwood, Cherie Drenzek, Aaron Fleischauer, Ashley Gent, Amanda Hartley, Liam Hicks, Mikhail Hoskins, Amanda Jara, Amanda Jones, Saadiah I. Khan, Ishrat Kamal-Ahmed, Sarah Kangas, FNU Kanishka, Alison Kleppinger, Anna Kocharian, Tomás M. León, Ruth Link-Gelles, B. Casey Lyons, John Masarik, Andrea May, Donald McCormick, Stephanie Meyer, Lauren Milroy, Keeley J. Morris, Lauren Nelson, Enaholo Omoike, Michael Pietrowski, Melissa A. Pike, Tamara Pilishvili, Xandy Peterson Pompa, Charles Powell, Kevin Praetorius, Adam Schiller, Mayra L. Smith-Coronado, Emma Stanislawski, Kyle Strand, Buddhi P. Tilakaratne, Hailey Vest, Caleb Wiedeman, Heather M. Scobie.

**Visualization:** Allison Zaldivar.

**Writing – original draft:** Maya R. Sternberg, Benjamin Silk, Heather M. Scobie.

**Writing – review & editing:** Maya R. Sternberg, Amelia Johnson, Justice King, Akilah R. Ali, Lauren Linde, Abiola O. Awofeso, Jodee S. Baker, Nagla S. Bayoumi, Steven Broadway,

Katherine Busen, Carolyn Chang, Iris Cheng, Mike Cima, Abi Collingwood, Vajeera Dora-bawila, Cherie Drenzek, Aaron Fleischauer, Ashley Gent, Amanda Hartley, Liam Hicks, Mikhail Hoskins, Amanda Jara, Amanda Jones, Saadiah I. Khan, Ishrat Kamal-Ahmed, Sarah Kangas, FNU Kanishka, Alison Kleppinger, Anna Kocharian, Tomás M. León, Ruth Link-Gelles, B. Casey Lyons, John Masarik, Andrea May, Donald McCormick, Stephanie Meyer, Lauren Milroy, Keeley J. Morris, Lauren Nelson, Enaholo Omoike, Komal Patel, Michael Pietrowski, Melissa A. Pike, Tamara Pilishvili, Xandy Peterson Pompa, Charles Powell, Kevin Praetorius, Eli Rosenberg, Adam Schiller, Mayra L. Smith-Coronado, Emma Stanislawski, Kyle Strand, Buddhi P. Tilakaratne, Hailey Vest, Caleb Wiedeman, Allison Zaldivar, Benjamin Silk, Heather M. Scobie.

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
