## [Decision Letter · Decision Letter 0]

13 Jun 2023

PONE-D-23-06056Application of a life table approach to assess duration of BNT162b2 vaccine-derived immunity by age using COVID-19 case surveillance data during the Omicron variant periodPLOS ONE

Dear Dr. Sternberg,

Thank you for submitting your manuscript to PLOS ONE. After careful consideration, we feel that it has merit but does not fully meet PLOS ONE’s publication criteria as it currently stands. Therefore, we invite you to submit a revised version of the manuscript that addresses the points raised during the review process.

We look forward to receiving your revised manuscript.

Kind regards,

Barbara T Rumain, PhD

Academic Editor

PLOS ONE

Journal Requirements:

Additional Editor Comments:

Reviewer #1:

This report described the application of a life table analysis using COVID-19 case surveillance and vaccine administration data to evaluate the differences in hazard rates among people vaccinated and unvaccinated over time. It showed the waning of vaccine protection against SARS-CoV-2 infection by age group during the Omicron period. In particular, it showed the largest decline of vaccine efficiency was observed among children aged 5–11 years and 12–17 years, while the declines among those 18 years and older were more modest after 19 weeks.

Major critique

1. A key assumption for the life table analysis is that the COVID-19 risk among subjects who are censored does not differ from the COVID-19 risk among those whose events are observed. As such, a seemly decreased hazard ratio of unvaccinated to the vaccinated could be due to increased risk among those with observed events. Therefore, one explanation of the seemly phenomenon of faster decay of VE in the young group can be that the deviation from the assumption grew larger along time for the young group than the older group.

Minor critique

2. The first paragraph of Materials & Methods: “from 22 U.S. jurisdictions … through two different mechanisms. Eight jurisdictions reported … 12 jurisdictions reported …”—Please clarify the separate descriptions of 8 and 12 jurisdictions corresponded to the stated “two different mechanisms” or not? If so, 8+12 equals to 20 instead of 22; if not so, please rephrase as it’s unclear and confusing here.

3. The four cohorts, January 16–February 5, February 6–26, February 27–April 2, and April 3–30, include 20, 20, 34, and 27 days respectively. What’s the rational to divide these four cohorts unevenly?

Reviewer #2:

Comment 1: Abstract Results are confusing.

In the abstract it is confusing and almost seems contradictory when you say:

• “The percent reduction in hazard rates for vaccinated compared with unvaccinated persons was lowest among children aged 5–11 years at 35.5% (95% CI: 33.3%, 37.6%) compared to the older adults, which ranged from 68.7%–89.6%.” but then

• “The largest declines were observed among children aged 5–11 years and 12–17 years, while the declines among those 18 years and older were more modest after 19 weeks.”

• Also, how is this last sentence: “Among those 18 and older, the percent reduction in hazard rates among vaccinated compared with unvaccinated ranged from 66%–81%.” different from the 68.7% to 89.6% you mentioned above?

I found these results confusing and seemingly contradicting.

Also would either report % decimals everywhere or nowhere and not mix.

Comment 2: I think this could be worded more clearly “differences in prior infection-derived immunity by vaccination status” … do you mean that unvaccinated people are more likely to derive infection-induced immunity over time? If so, I would clarify as the language is non-specific.

Comment 3: Please cite Tartof et al. in Lancet for this sentence [as well], which was really the first paper that outlined this: “A study during the Delta period showed evidence of waning protection against infection but strong protection against death 6 months after vaccination”

Comment 4: Are results similar if you excluded all the sites where you had to make the unvaccinated correction factor? May be a worthwhile sensitivity analysis perform and at least mention that this did not majorly influence the results, even if only about 5% of overall data.

Comment 5: Did you test this key assumption in any way? Can you reassure the reader this is probably not a major issue? “A key assumption for the life table analysis is that the COVID-19 risk among subjects who are 191 censored does not differ from the COVID-19 risk among those whose events are observed (known as non 192 informative censoring) [15].”

Comment 6: “Since there are no seroprevalence data available by vaccination status, we assumed that infections preferentially occurred in unvaccinated people during periods after the vaccine had been authorized.” This is probably true, and your method of handling this was very clever, but did you model the impact if you assumed equal likelihood of infection between the two groups? This would model a world if vaccine did not significantly interrupt transmission or infection, but lessened severity instead. Would be an important sensitivity analysis perhaps.

Comment 7: How do explain the potential negative reductions in 5-11 year-olds in Table 2? This will probably be picked up by several readers?

Comment 8: This entire analysis is based on cases. How do you deal with differences in healthcare seeking / testing behavior which may be a major factor, especially for the endpoint of COVID-19 cases? Can you perform this same analysis on hospitalization?

Comment 9: What does the red dashed line in figure 4 at 80% represent?

Comment 10: Did school closures during the study period potential impact any of the findings? Is there any evidence that 5-11 year-olds kept more social interaction through daycare/after school programs compared to 12-17 year olds who could likely stay home on their own? Perhaps an explanation of the 5-11 year old findings? Are there other larger social influences besides the lower dose that could explain this? Might be worth contextualizing.

Comment 11: I mentioned before the impact of testing proclivity. You mention very clearly in the results that “We were unable to account for differences in underlying conditions, testing practices, prior infection, or prevention behaviors by vaccination status, age, geography, and over time; current results were influenced by the circulating Omicron variant, the relatively high prevalence of prior infection among children, and differences in testing practices and vaccine seeking behaviors compared to 2021.” I guess the question many will ask, is—if so many limitations of this life-table approach—why even perform it? Doesn’t it have way more limitations than adjusted analyses or TND approaches that can help condition on testing? I think this should be better explained by the authors. What does this approach add? Don’t we know vaccines wane after about 2-3 months against mild illness already? What does this approach and paper clearly add? I would spell this out a little better—because it just feels like a crude analysis when others have already reported better adjusted results.

Comment 12: Finally, although the vaccine does wane, with the exception of the 5-11 group… it seems like the vaccine continues to provide protection through week 19 and likely beyond. Is this consistent with TNDs that have looked at VE against infection/symptomatic disease beyond 3 or 4 months? How do you characterize this? I would highlight you still see protection beyond 19 weeks in this study… despite some waning—because that is what you find. But contextualize with other reports.

PLOS Data Policy: Have the authors made all data underlying the findings in their manuscript fully available? No

Reviewers' comments:

Reviewer's Responses to Questions

**Comments to the Author**

1. Is the manuscript technically sound, and do the data support the conclusions?

Reviewer #1: Partly

Reviewer #2: Partly

2. Has the statistical analysis been performed appropriately and rigorously? 

Reviewer #1: Yes

Reviewer #2: Yes

3. Have the authors made all data underlying the findings in their manuscript fully available?

Reviewer #1: No

Reviewer #2: Yes

4. Is the manuscript presented in an intelligible fashion and written in standard English?

Reviewer #1: Yes

Reviewer #2: Yes

5. Review Comments to the Author

Reviewer #1: This report described the application of a life table analysis using COVID-19 case surveillance and vaccine administration data to evaluate the differences in hazard rates among people vaccinated and unvaccinated over time. It showed the waning of vaccine protection against SARS-CoV-2 infection by age group during the Omicron period. In particular, it showed the largest decline of vaccine efficiency was observed among children aged 5–11 years and 12–17 years, while the declines among those 18 years and older were more modest after 19 weeks.

Major critique

1. A key assumption for the life table analysis is that the COVID-19 risk among subjects who are censored does not differ from the COVID-19 risk among those whose events are observed. As such, a seemly decreased hazard ratio of unvaccinated to the vaccinated could be due to increased risk among those with observed events. Therefore, one explanation of the seemly phenomenon of faster decay of VE in the young group can be that the deviation from the assumption grew larger along time for the young group than the older group.

Minor critique

2. The first paragraph of Materials & Methods: “from 22 U.S. jurisdictions … through two different mechanisms. Eight jurisdictions reported … 12 jurisdictions reported …”—Please clarify the separate descriptions of 8 and 12 jurisdictions corresponded to the stated “two different mechanisms” or not? If so, 8+12 equals to 20 instead of 22; if not so, please rephrase as it’s unclear and confusing here.

3. The four cohorts, January 16–February 5, February 6–26, February 27–April 2, and April 3–30, include 20, 20, 34, and 27 days respectively. What’s the rational to divide these four cohorts unevenly?

Reviewer #2: Comment 1: Abstract Results are confusing.

In the abstract it is confusing and almost seems contradictory when you say:

• “The percent reduction in hazard rates for vaccinated compared with unvaccinated persons was lowest among children aged 5–11 years at 35.5% (95% CI: 33.3%, 37.6%) compared to the older adults, which ranged from 68.7%–89.6%.” but then

• “The largest declines were observed among children aged 5–11 years and 12–17 years, while the declines among those 18 years and older were more modest after 19 weeks.”

• Also, how is this last sentence: “Among those 18 and older, the percent reduction in hazard rates among vaccinated compared with unvaccinated ranged from 66%–81%.” different from the 68.7% to 89.6% you mentioned above?

I found these results confusing and seemingly contradicting.

Also would either report % decimals everywhere or nowhere and not mix.

Comment 2: I think this could be worded more clearly “differences in prior infection-derived immunity by vaccination status” … do you mean that unvaccinated people are more likely to derive infection-induced immunity over time? If so, I would clarify as the language is non-specific.

Comment 3: Please cite Tartof et al. in Lancet for this sentence [as well], which was really the first paper that outlined this: “A study during the Delta period showed evidence of waning protection against infection but strong protection against death 6 months after vaccination”

Comment 4: Are results similar if you excluded all the sites where you had to make the unvaccinated correction factor? May be a worthwhile sensitivity analysis perform and at least mention that this did not majorly influence the results, even if only about 5% of overall data.

Comment 5: Did you test this key assumption in any way? Can you reassure the reader this is probably not a major issue? “A key assumption for the life table analysis is that the COVID-19 risk among subjects who are 191 censored does not differ from the COVID-19 risk among those whose events are observed (known as non 192 informative censoring) [15].”

Comment 6: “Since there are no seroprevalence data available by vaccination status, we assumed that infections preferentially occurred in unvaccinated people during periods after the vaccine had been authorized.” This is probably true, and your method of handling this was very clever, but did you model the impact if you assumed equal likelihood of infection between the two groups? This would model a world if vaccine did not significantly interrupt transmission or infection, but lessened severity instead. Would be an important sensitivity analysis perhaps.

Comment 7: How do explain the potential negative reductions in 5-11 year-olds in Table 2? This will probably be picked up by several readers?

Comment 8: This entire analysis is based on cases. How do you deal with differences in healthcare seeking / testing behavior which may be a major factor, especially for the endpoint of COVID-19 cases? Can you perform this same analysis on hospitalization?

Comment 9: What does the red dashed line in figure 4 at 80% represent?

Comment 10: Did school closures during the study period potential impact any of the findings? Is there any evidence that 5-11 year-olds kept more social interaction through daycare/after school programs compared to 12-17 year olds who could likely stay home on their own? Perhaps an explanation of the 5-11 year old findings? Are there other larger social influences besides the lower dose that could explain this? Might be worth contextualizing.

Comment 11: I mentioned before the impact of testing proclivity. You mention very clearly in the results that “We were unable to account for differences in underlying conditions, testing practices, prior infection, or prevention behaviors by vaccination status, age, geography, and over time; current results were influenced by the circulating Omicron variant, the relatively high prevalence of prior infection among children, and differences in testing practices and vaccine seeking behaviors compared to 2021.” I guess the question many will ask, is—if so many limitations of this life-table approach—why even perform it? Doesn’t it have way more limitations than adjusted analyses or TND approaches that can help condition on testing? I think this should be better explained by the authors. What does this approach add? Don’t we know vaccines wane after about 2-3 months against mild illness already? What does this approach and paper clearly add? I would spell this out a little better—because it just feels like a crude analysis when others have already reported better adjusted results.

Comment 12: Finally, although the vaccine does wane, with the exception of the 5-11 group… it seems like the vaccine continues to provide protection through week 19 and likely beyond. Is this consistent with TNDs that have looked at VE against infection/symptomatic disease beyond 3 or 4 months? How do you characterize this? I would highlight you still see protection beyond 19 weeks in this study… despite some waning—because that is what you find. But contextualize with other reports.

6. PLOS authors have the option to publish the peer review history of their article (what does this mean?). If published, this will include your full peer review and any attached files.

Reviewer #1: No

Reviewer #2: **Yes: **John M McLaughlin, PhD

---

## [Author Response · Author response to Decision Letter 0]

2 Aug 2023

We thank the reviewers for the helpful comments. Please note that the line numbers indicated in our letter to the response to the reviewers refers to the unmarked manuscript.

---

## [Editor Report · Decision Letter 1]

4 Sep 2023

Application of a life table approach to assess duration of BNT162b2 vaccine-derived immunity by age using COVID-19 case surveillance data during the Omicron variant period

PONE-D-23-06056R1

Dear Dr. Sternberg,

We’re pleased to inform you that your manuscript has been judged scientifically suitable for publication and will be formally accepted for publication once it meets all outstanding technical requirements.

Kind regards,

Barbara T Rumain, PhD

Academic Editor

PLOS ONE

---

## [Editor Report · Acceptance letter]

12 Sep 2023

PONE-D-23-06056R1 

­­Application of a life table approach to assess duration of BNT162b2 vaccine-derived immunity by age using COVID-19 case surveillance data during the Omicron variant period 

Dear Dr. Sternberg:

I'm pleased to inform you that your manuscript has been deemed suitable for publication in PLOS ONE. Congratulations! Your manuscript is now with our production department. 

Kind regards, 

on behalf of

Dr. Barbara T Rumain 

Academic Editor

PLOS ONE